# Fulfillment, burnout and resilience in emergency medicine—Correlations and effects on patient and provider outcomes

**Revathi Jyothindran, James P. d'Etienne, Kevin Marcum, Aubre Tijerina, Clare Graca, Heidi Knowles, Bharti R. Chaudhari, Nestor R. Zenarosa, Hao Wang**[ID] *

Department of Emergency Medicine, Integrative Emergency Services, Dallas, Texas, United States of America

* hwang@ies.healthcare

## Abstract

### Background

Healthcare provider wellness have been reported to correlate with patient care outcomes. It is not understood whether synergistic effects may exist between them.

### Objective

We aim to investigate three provider wellness markers and determine their associations with provider self-reported medical errors and intent-to-leave outcomes among Emergency Department (ED) providers.

### Design

This is a multi-center retrospective study.

### Method

Three wellness domains include professional fulfillment (PF), burnout (BO), and personal resilience (PR). Two outcomes measured as provider self-reported medical errors and provider intent-to-leave. Correlations between wellness markers and outcomes were analyzed. When adjusted for other confounders (provider demographics, provider experience, and operational environment), a multivariate logistic regression analysis was performed to further determine the interactions among these three domains on provider wellness affecting patient and provider related outcomes.

### Results

Total 242 surveys were collected from providers at 16 different EDs. The median score of PF were 2.83 among physicians and 2.67 among APPs, BO were 1.00 (physicians) and 0.95 (APPs), and PR were 0.88 (physicians) and 0.81 (APPs). The median scores of self-reported medical errors were 1.50 (physicians) and 0.95 (APPs), and intent-to-leave were 1.00 (physicians and APPs). High correlations occurred among PF, BO, and PR. When

**Data Availability Statement:** Data cannot be shared publicly because of special contract agreement with the third party. The authors of this study had no special access privileges in accessing

the datasets which other interested researchers would not have. However, data might be available from the Integrative Emergency Service Data Access upon request to the Director of Physician Wellness, Dr. Jyothindran via jyothindran@bswhealth.org.

**Funding:** The authors received no specific funding for this work.

**Competing interests:** The authors have declared that no competing interests exist.

analyzed together, high PF, low BO, and high PR functioned as a protective effect on provider intent-to-leave (adjusted odds ratios = 0.09, 95% CI 0.03–0.30).

## Conclusion

High correlations occurred among three provider wellness markers with no significant difference between physicians and APPs. Providers with high PR, low BO, and high PR tended to be more stable in their jobs.

## Introduction

Healthcare provider wellness has been recognized as a factor that plays an important role in patient-centered care in recent years [1–3]. Previous studies have shown that physician wellness was positively associated with patient satisfaction, decreased medical errors, and improved patient clinical outcomes [1, 4–6]. Professional fulfillment (PF) is defined as happiness or meaningfulness, self-worth, self-efficacy and satisfaction at work [7]. Studies focusing on PF and professional satisfaction have demonstrated that physicians with high PF or satisfaction are associated with higher patient satisfaction [8, 9]. Whereas, ones with lower PF correlated with increased medical errors and higher professional burnout (BO),—an emotional exhaustion, depersonalizations, and reduced feelings of personal accomplishment [4, 10]. On the other hand, personal resilience (PR)–the capacity to respond to stress in a healthy way–has been studied using sleep-related impairment, provider anxiety and depression measures [11–13]. Providers with higher PR tended to be more sustainable in the health care workforce [11]. Promoting PR could potentially decrease provider BO and increase the quality of care while reducing medical errors [12, 14–16].

In terms of these provider wellness domains, previous studies have focused on specialties of family or internal medicine with few reported in the field of Emergency Medicine (EM). Given a common belief that EM providers have higher rates of stress, anxiety, and burnout [17, 18], it is important to investigate wellness conditions using similar tools amongst Emergency Department (ED) providers. Furthermore, provider wellness markers may have moderating effects on each other or interactions among different domains that could affect outcome measures. One domain functioning as a moderator can further affect other domains functioned as the mediators, so that the mediators and moderators would interact in the same model [19]. Therefore, it is necessary to determine whether provider wellness markers have internal correlation with potential interactions that could positively or negatively affect outcomes.

At present, interactions among healthcare provider wellness domains have rarely been reported, especially regarding their association with either patient care outcomes (i.e. medical errors) or provider related outcomes (i.e. job retention). We hypothesized that the unique environment in which ED providers practice differentiates them from other specialties, wellness conditions differ from other specialties, and both patient safety and provider job security may be unique to that environment. Therefore, to better understand ED provider wellness conditions and their relationship to patient and provider outcomes, we aim to: 1) measure ED provider wellness domains, specifically PF, BO, and PR; 2) determine the correlation of these wellness markers to patient-related medical errors and provider intent-to-leave; and 3) further investigate interactions among all three wellness domains and its association with the study outcomes.

## Methods

### Study design

This was a secondary data analysis of a previous quality improvement project focusing on healthcare provider wellness within Integrative Emergency Services (IES) group, an Emergency Medicine (EM) group providing EM coverage of 16 hospital EDs mainly in the State of Texas, USA. This project was intended to include all ED providers within IES group. Therefore, the survey was sent to all ED providers including physicians and Advanced Practice Providers (APP). APP includes physician assistant (PA) and nurse practitioner (NP). Data were collected prospectively online using Qualtrics Survey Software (Provo, UT) from January to March 2018, via the Stanford Wellness Survey, created by the WellMD team at Stanford University. The survey questionnaires were provided by the Stanford WellMd Center with contractual agreement and permission to use data in secondary analysis. Due to the nature of secondary data analysis with deidentified personal information, this study was waived for approval by the local Institutional Review Board (IRB).

### Provider wellness survey

The study provider wellness survey was reported previously in the literature and was a closed survey. Survey was sent to all ED providers via email with a link to the survey website. The first two pages of the survey included the general instruction of survey completion and informed consent. Survey was not proceeded if providers declined to participate. Survey website were active, and providers were allowed to enter multiple times during the entire study period. The participation of this project was totally volunteer with no incentive provided regardless of the participations. While answering the survey, providers had opportunity to review their response using the back button and were able to change their response before the final submission. We identified unique participant based on the general information and email/IP address that individual participant provided. We allowed multiple entrances of the survey during the study period. If duplicated surveys were found, the latest one was used for the final analysis.

### Study setting and participants

This study enrolled healthcare providers from 16 different hospital EDs across the state of Texas, USA. Among all 16 EDs, 2 EDs have extremely high annual volume (>100,000/year), 5 EDs have moderate to high annual volume (60,000–100,000/year), while the other 9 EDs have low to moderate annual volume (<60,000/year). We included surveys from qualified ED healthcare providers who agreed to participate in this study. We excluded surveys: 1) from providers who declined to participate; 2) empty surveys; 3) incomplete surveys (<10% of completions); and 4) duplicate surveys.

### Provider wellness measurements

Three areas of healthcare provider wellness were measured including professional fulfillment (PF), burnout (BO), and personal resilience (PR). PF and BO were measured using a 16-point professional fulfillment index (PFI) questionnaire with good reliability [4]. PFI includes measurement within three domains (professional fulfillment, emotional exhaustion, and interpersonal disengagement) covering two distinct areas (PF and BO). PR measures sleep-related impairment, provider anxiety, and provider depression. Sleep-related impairment was measured using an 8-item questionnaire, a short version of the Pittsburgh Sleep Quality Index and portion of the Patient Reported Outcomes Measurement Information System (PROMIS) tool [4, 13, 20]. Provider anxiety and depression measurement included a 4-item questionnaire

which was also derived from the PROMIS tool [4, 21]. Each item was scored on a five-point Likert scale (0 to 4). The overall score of each area was calculated by averaging the total item scores. High scores indicated providers had high PF, high BO, or low PR (defined as high sleep-related impairment, high anxiety, and high depression levels). These provider wellness measures have all been previously reported and validated [4, 20, 21].

## Outcome measurements

We measured two outcomes (patient-related and provider-related). Provider self-reported medical errors were considered patient-related outcome measures. We used 4-item question-naire tool that was previously reported in the literature for medical error measurements [4]. Briefly, medical errors were classified as errors resulting in patient harm (include any adverse events occurred linked to the medical errors), wrong medication (incorrected medications including names, doses, and type of administrations, that had administered to patients, or had not given to the patients but recognized by others (pharmacists, nurses, or other providers), or wrong lab test (any incorrected lab tests initially ordered in the computerized physician order entry (CPOE) system regardless of the completions), etc. that occurred both recently (e.g., the previous week) and within providers' working lifetime on a six-point Likert scale ranging 0–5. The overall score was calculated by averaging the total item scores. Provider intent-to-leave was considered provider-related outcome measurement. This was a one-item question asking the providers' likelihood of leaving the institution within two years using a 5-point Likert scale ranging from 0 (None) to 4 (definitely), that was also reported in the literature. The investigators decided a priori using Delphi technique that scores of <1 for intent-to-leave and <2 for self-reported medical errors were considered "low" scores [22, 23].

## Study variables

Provider demographics included sex, race, ethnicity, and age. Other variables included volume of primary practice ED, years in practice at the practice site, and years since completion of training. We also surveyed whether providers lived with a significant other and/or children.

## Data analysis

We intended to measure ED provider wellness markers in three domains: professional fulfill-ment, burnout, and personal resilience. We used Cronbach's alpha (α) to determine internal consistency of professional fulfillment, burnout, and personal resilience measures. An α>0.8 was considered good reliability and α>0.7 was considered adequate reliability. Skewness and kurtosis were used to determine whether wellness markers were normally distributed. | Skewness|<0.5 was mildly to normally distributed, 1>|Skewness|≥0.5 was moderately skewed, and |Skewness|≥1 was highly skewed. Kurtosis>3 was considered data less normally distrib-uted. Furthermore, we investigated the associations between provider wellness markers and outcomes. We initially used correlation co-efficiency (r) with |r|≥0.5 indicating strong correla-tions, 0.5>|r|≥0.3 indicating moderate correlations, and 0.3>|r|≥0.1 indicating weak correla-tions. Secondly, we performed a multivariate logistic regression to determine the association between different provider wellness markers and study outcomes (self-reported medical errors, and provider intent-to-leave). After completed appropriate diagnostic checks to iden-tify the outliers and determine the collinearity, all available independent variables were initially included in the regression model. Variables including provider demographics and practice environment, in addition to measured wellness markers, were analyzed as potential indepen-dent risk predictors of patient-related and provider-related outcomes. The backward stepwise variable selection was applied to obtain the final regression model. Significance levels for entry

and to stay were set at 0.1 to avoid exclusion of potential candidate variables. The final regression model was determined by sequentially excluding individual variables with a p-value > 0.05 until all regression coefficients were significant. The overall model performance was tested using the Hosmer–Lemeshow goodness-of-fit test. Finally, interaction analysis was performed to determine whether these three wellness markers have interactions which affect study outcomes. To quantify such interactions, investigators use previously reported cut off of >3 for high "professional fulfillment" and 1.33 for high "burnout" scores [4]. Low PR was defined as above 1.25 based on the upper quartile scores of the entire cohort. All analyses were performed using Stata v14.2 (College Station, Texas).

## Results

From January to March 2018, a total of 382 surveys were sent out to all ED providers. We received 326 surveys with over 85% (326/382) of response rate. A final completion rate of 89% (289/326) was yielded after the exclusion of 37 providers who declined to participate. Providers worked at 16 different hospital EDs, included full-licensed ED physicians and APPs (including physician assistants and nurse practitioners). Among all received surveys, we also excluded 47 invalid surveys (31 empty surveys and 16 surveys with <10% of completions) with 242 surveys placed in a final analysis (see detail in Fig 1).

Table 1 shows the general characteristics of the study participants. Our study included 146 ED physicians and 96 APPs. Males were predominant among ED physician and females were predominant among APP participants. Most participants were White, non-Hispanic, and practicing at moderate to high volume EDs. Over half of the participants were out of training less than 10 years and have been practicing at designated ED less than 5 years (see Table 1).

Provider wellness markers including PF, BO, and PR were measured and descriptive analyses are reported in Table 2. Data from most wellness markers were mildly skewed. Therefore,

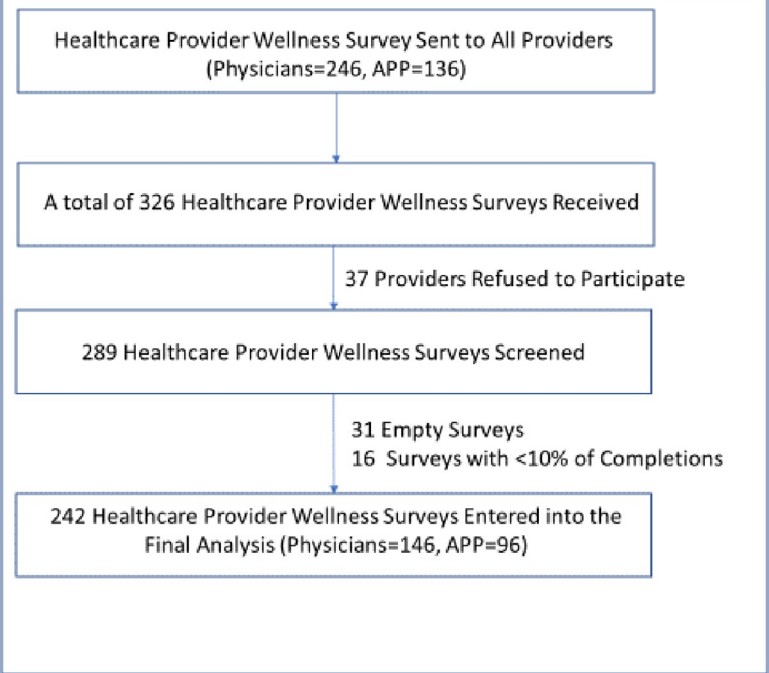

**Fig 1. Study flow diagram.**

**Table 1. General characteristics of study participants.**

| | Physician (N = 146) | APP (N = 96) |
|---|---|---|
| Gender---n (%) | | |
| Male | 101 (69) | 34 (35) |
| Female | 43 (29) | 62 (65) |
| Unknown | 2 (1.4) | |
| Age---n (%) | | |
| <30 years old | 2 (1.4) | 9 (9.4) |
| 30–39 years old | 75 (51) | 44 (46) |
| 40–49 years old | 37 (25) | 23 (24) |
| 50–59 years old | 19 (13) | 18 (19) |
| ≥60 years old | 10 (6.9) | 2 (2.1) |
| Unknown | 3 (2.1) | |
| Race---n (%) | | |
| White | 108 (74) | 75 (78) |
| Black or African American | 5 (3.4) | 4 (4.2) |
| Asian | 28 (19) | 13 (14) |
| Others* | 5 (3.4) | 4 (4.2) |
| Ethnicity---n (%) | | |
| Hispanic/Latino | 4 (2.7) | 4 (4.2) |
| Not Hispanic/Latino | 142 (97) | 91 (95) |
| Unknown | | 1 (1.0) |
| Participant Primary Hospital ED Size---n (%) | | |
| Low ED Annual Volume (<60,000/year) | 18 (12) | 18 (19) |
| Moderate ED Annual Volume (60,000–100,000/year) | 59 (40) | 44 (46) |
| High ED Annual Volume (>100,000/year) | 66 (45) | 30 (31) |
| Unknown | 3 (2.1) | 4 (4.2) |
| Practice Years out from Training---n (%) | | |
| <5 years | 51 (35) | 38 (40) |
| 5–10 years | 44 (30) | 31 (32) |
| 11–15 years | 20 (14) | 13 (14) |
| 16–20 years | 10 (6.9) | 9 (9.4) |
| >20 years | 21 (14) | 4 (4.2) |
| Unknown | | 1 (1.0) |
| Practice Years at Primary Participant ED---n (%) | | |
| 0–5 years | 96 (66) | 62 (65) |
| 6–10 years | 28 (19) | 24 (25) |
| 11–15 years | 9 (6.2) | 9 (9.4) |
| >15 years | 13 (8.9) | 1 (1.0) |
| Relationship---n (%) | | |
| Not living with a significant other | 15 (10) | 25 (26) |
| Living with a significant other | 129 (88) | 70 (73) |
| Not living with dependent children | 51 (35) | 42 (44) |
| Living with dependent children | 94 (64) | 53 (55) |

Other* includes American Indian or Alaska Native, Native Hawaiian or Pacific Islander, or Unknown.

both mean with standard deviation (SD) and median with interquartile range (IQR) were reported. Internal consistency was measured among different wellness markers using

**Table 2. Descriptive analysis of healthcare provider wellness survey and outcome measurements.**

| Measurement | Physicians | APPs | P value |
|---|---|---|---|
| Healthcare Provider Wellness Measurements | | | |
| Professional Fulfillment (PF) | | | |
| Mean (SD) | 2.71(0.78) | 2.64(0.77) | 0.47 |
| Median (IQR) | 2.83(2.17,3.33) | 2.67 (2.17,3.08) | 0.54 |
| Skewness/Kurtosis/Cronbach α | -0.39/2.76/0.91 | -0.60/3.43/0.89 | |
| Burnout (BO) | | | |
| Mean (SD) | 1.06(0.69) | 1.05(0.72) | 0.88 |
| Median (IQR) | 1.00(0.60,1.40) | 0.95 (0.50, 1.50) | 0.87 |
| Skewness/Kurtosis/Cronbach α | 0.95/4.24/0.94 | 0.56/2.90/0.92 | |
| Personal Resilience (PR) | | | |
| Mean (SD) | 0.96(0.51) | 0.93(0.60) | 0.68 |
| Median (IQR) | 0.88(0.56,1.25) | 0.81(0.44,1.31) | 0.36 |
| Skewness/Kurtosis/Cronbach α | 0.80/3.25/0.90 | 0.87/3.49/0.92 | |
| Outcome Measurements | | | |
| Patient-related (Self-reported Medical Errors) | | | |
| Mean (SD) | 1.63(0.86) | 1.08(0.68) | <0.001 |
| Median (IQR) | 1.50(1.00,2.25) | 1.00(0.50,1.50) | <0.001 |
| Skewness/Kurtosis | 0.25/2.46 | 0.37/2.90 | |
| Provider-related (Intent-to-leave) | | | |
| Mean (SD) | 0.82(1.00) | 0.96(1.01) | 0.31 |
| Median (IQR) | 1.00(0,1.00) | 1.00(0,2.00) | 0.22 |
| Skewness/Kurtosis | 1.27/4.24 | 0.96/3.38 | |

Cronbach's α, all showed good internal consistency (see detail in Table 2). No significant differences in terms of PF, BO, and PR were found between physician and APP groups. However, physicians tended to report more medical errors than those of the APP's (p<0.001).

Table 3 shows correlations between different wellness markers. Moderate to high correlations were found in PF, BO, and PR regardless of types of ED providers. Regarding outcome measurements, provider wellness markers seem to have no correlation to self-reported medical errors among physicians but showed weak to moderate correlation with APPs. Additionally, better correlations were found between wellness markers and provider intent-to-leave in physicians than in APPs.

**Table 3. Correlations between different healthcare provider wellness markers and outcomes.**

| | **Professional Fulfillment** | **Burnout** | **Personal Resilience** | **Medical Errors** |
|---|---|---|---|---|
| Physicians | | | | |
| Professional Fulfillment | X | | | |
| Burnout | -0.61 | X | | |
| Personal Resilience | -0.48 | 0.59 | X | |
| Self-reported Medical Errors | -0.07 | 0.09 | 0.06 | X |
| Intent-to-Leave | -0.50 | 0.44 | 0.35 | 0.05 |
| APPs | | | | |
| Professional Fulfillment | X | | | |
| Burnout | -0.58 | X | | |
| Personal Resilience | -0.49 | 0.55 | X | |
| Self-reported Medical Errors | -0.27 | 0.44 | 0.32 | X |
| Intent-to-Leave | -0.36 | 0.43 | 0.19 | 0.14 |

**Table 4. Different wellness markers affecting patient and provider related outcomes.**

| | Adjusted Odds Ratio | 95% Confidence Interval | P value |
|---|---|---|---|
| No Wellness Markers Affecting Provider Self-Reported Medical Errors | | | |
| Provider | | | |
| ED Physicians | Reference | Reference | Reference |
| ED APPs | 0.34 | 0.21–0.56 | <0.001 |
| Two Wellness Markers Affecting Provider Intent-to-Leave Outcomes | | | |
| Professional Fulfillment (PF) | | | |
| Low | Reference | Reference | Reference |
| High | 0.22 | 0.09–0.57 | 0.002 |
| Burnout (BO) | | | |
| High | Reference | Reference | Reference |
| Low | 0.39 | 0.19–0.76 | 0.006 |

To further determine associations between different wellness markers and outcomes, a multivariate logistic regression analysis was performed to analyze the risk of wellness markers affecting either patient-related or provider-related outcomes. Table 4 shows no such effects of wellness markers on provider self-reported medical errors, consistent with no correlation shown in Table 3. When provider wellness markers were analyzed to determine the association with provider intent-to-leave, high PF and low BO seemed to independently correlate with low provider intent-to-leave scores. Hosmer–Lemeshow goodness-of-fit test showed no statistically significant difference (p>0.05) indicating data fit the model well. When all three wellness markers were analyzed together, individual markers seemed to have fewer protective effects on provider intent-to-leave. Table 5 demonstrates the significant protective effects when high PF, low BO, and high PR were combined (p<0.001).

## Discussion

In this study, we found that ED physicians and APPs had similar professional fulfillment, burnout, and personal resilience scores. Moderate-to-high correlations were also found among these wellness domains in both physician and APP groups. They correlated better with provider intent-to-leave than self-reported medical errors. When all three wellness domains were analyzed together, it was noted that high professional fulfillment, low burnout, and high personal resilience tended to have a protective effect related to intent to leave current position. Our study findings link wellness domains to patient and provider outcomes and these findings provide important information to help future ED provider wellness programs improve quality and patient-centered care.

Clinicians, in general, tend to have higher burnout than other professions [24]. It is postulated that ED providers in particular should be investigated as a specialty differing from others

**Table 5. Interactions among different wellness markers affecting provider intent-to-leave outcomes.**

| | Adjusted Odds Ratio | 95% Confidence Interval | P value |
|---|---|---|---|
| Interactions among PF, BO, and PR | | | |
| Low PF (↓), High BO (↑), Low PR (↓) (↓PF↑BO↓PR) | Reference | Reference | Reference |
| ↓PF↑BO↑PR | 1.49 | 0.43–5.19 | 0.535 |
| ↓PF↓BO↓PR | 0.73 | 0.20–2.69 | 0.641 |
| ↑PF↑BO↓PR | 6.58 | 0.17–251.8 | 0.311 |
| ↑PF↓BO↑PR | 0.09 | 0.03–0.30 | <0.001 |

due to the nature of rotating shift schedules and stressful work environment. ED providers face challenges of high patient volume and varying severity of illness, often high, which may contribute to levels of stress. Given similar working conditions, this might result in similar burnout levels between ED physicians and APPs. However, due to different responsibility of physicians and APPs, their self-reported medical errors might be different. In study EDs, physicians play supervising roles on APPs and answer any questions that APPs may ask though physicians might not see every APPs' patients during their shift. The challenges of working rotating shift schedules (mornings, evenings, and nights) may also affect ED provider personal resilience. We found that older providers who tended to work fewer night shifts (due to group option to reduce or eliminate nights based on age) had higher personal resilience (low score) when compared to providers from younger groups (see S1 Table).

When each wellness domain was measured, it showed high internal consistencies similar to previous studies [4, 25, 26]. Since there are no similar report in the literature, we provide initial wellness measurements using the average scores in each domain and further divided scores into higher and lower categories. Our findings to determine wellness domain cutoffs were based on a previous study [4]. Since the previous study did not focus on Emergency Medicine (EM) wellness measurements (included only <1% of EM healthcare providers) [4], such cutoffs might not be accurate for ED providers. Therefore, correlations and interactions among different wellness domains and its association with patient/provider centered outcomes might be more important than simply determining cutoff levels of provider wellness.

While literature reports of physician wellness have shown correlation with medical errors [27–29], our findings showed no correlations between provider burnout and medical errors. Medical errors which occur in healthcare systems usually happen due to a chain of failures, the actions of a single provider may not fully account for this, as they only represent a portion of the medical error chain. Like the Swiss cheese model, the occurrence of medical errors usually happens due to a failure of the whole healthcare system, as opposed to the individual [30]. In addition, these findings may indicate an offset effect when providers have opposite conditions in different domains, under reporting, or decreased recollection of those events.

When intent-to-leave was measured, however, poor physician wellbeing correlated better with high intent-to-leave, consistent with previous reports [31, 32]. Intent-to-leave might be more controllable by the provider themselves; therefore, poor physician wellbeing might play an important role in provider willingness to remain at current job. Additionally, we did find synergistic effects occurred regarding provider intent-to-leave. There was significant protection if providers ranked high in professional fulfillment, low in burnout, and high in personal resilience (Table 5). Wellness domains are not isolated within any given provider. The potential exists that provider wellness conditions at different domains have either offset or synergistic effects. Therefore, these domains should not be analyzed in isolation.

This study has its limitations. Firstly, provider wellness includes numerous areas and this study only chose three domains for provider wellness measurements, which are limited. More provider wellness domains should be studied in the future. Secondly, at present, all the provider wellness measurements in this study were self-reported, subjective, and perhaps less accurate. Adding physiological or biomarkers to future wellness investigations, in addition to surveys, may produce valuable information. Thirdly, study outcome measurements were limited to provider intent-to-leave and self-reported medical errors, which again, is subjective. Fourthly, provider wellness conditions are multi-factorial, by only analyzing provider demographics, years of practice, and relations to significant others and children in this study, similar to other studies, examines only a portion of factors effecting wellness. Other confounders, which could potentially affect study results, were not analyzed in this study. Lastly, although this is the largest study to date investigating these wellness markers specific to ED providers

our study sample was limited. The average of EM physicians in US is male, 40–50 age, with nearly 10 years of practicing in ED with moderate volume after the residency graduation [33]. However, our providers tend to be younger with less years of practice in the ED, which has some differences in comparison to national average. Therefore, our finding might lack of generalizability and require external validations. A large-scaled prospective multicenter study is warranted in the future to 1) measure provider wellness using both survey and biological markers; and 2) better and more accurately determine wellness conditions and its associations to patient/provider quality healthcare.

## Conclusion

High correlations occurred among three different provider wellness markers in ED healthcare providers with no significant difference between physicians and advanced practice providers. Providers with higher professional fulfillment, lower burnout, and higher personal resilience tended to report lower likelihood to leave their current jobs.

## Supporting information

**S1 Table. Personal resilience from healthcare providers of different age groups.** (DOCX)

## Author Contributions

**Conceptualization:** Revathi Jyothindran, James P. d'Etienne, Nestor R. Zenarosa, Hao Wang.

**Data curation:** Revathi Jyothindran, Kevin Marcum, Clare Graca, Hao Wang.

**Formal analysis:** Bharti R. Chaudhari, Hao Wang.

**Investigation:** James P. d'Etienne, Kevin Marcum, Aubre Tijerina, Clare Graca, Heidi Knowles, Bharti R. Chaudhari, Nestor R. Zenarosa, Hao Wang.

**Methodology:** Heidi Knowles, Bharti R. Chaudhari, Nestor R. Zenarosa, Hao Wang.

**Project administration:** Kevin Marcum, Aubre Tijerina, Clare Graca, Heidi Knowles.

**Resources:** Revathi Jyothindran, James P. d'Etienne, Aubre Tijerina, Heidi Knowles.

**Supervision:** Hao Wang.

**Validation:** Revathi Jyothindran, James P. d'Etienne, Kevin Marcum, Aubre Tijerina, Hao Wang.

**Writing – original draft:** Revathi Jyothindran, James P. d'Etienne, Hao Wang.

**Writing – review & editing:** Revathi Jyothindran, James P. d'Etienne, Kevin Marcum, Aubre Tijerina, Clare Graca, Heidi Knowles, Bharti R. Chaudhari, Nestor R. Zenarosa, Hao Wang.

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
