## [Decision Letter · Decision Letter 0]

19 Jun 2020

PONE-D-20-11949

Fulfillment, Burnout and Resilience in Emergency Medicine– Correlations and Effects on Patient and Provider Outcomes

PLOS ONE

Dear Dr. Wang,

Thank you for submitting your manuscript to PLOS ONE. After careful consideration, we feel that it has merit but does not fully meet PLOS ONE’s publication criteria as it currently stands. Therefore, we invite you to submit a revised version of the manuscript that addresses the points raised during the review process.

We look forward to receiving your revised manuscript.

Kind regards,

Andrew Carl Miller

Academic Editor

PLOS ONE

Journal Requirements:

3. Please include your tables as part of your main manuscript and remove the individual files. Please note that supplementary tables should be uploaded as separate "supporting information" files.

Reviewers' comments:

Reviewer's Responses to Questions

**Comments to the Author**

1. Is the manuscript technically sound, and do the data support the conclusions?

Reviewer #1: Yes

Reviewer #2: Yes

2. Has the statistical analysis been performed appropriately and rigorously? 

Reviewer #1: I Don't Know

Reviewer #2: I Don't Know

3. Have the authors made all data underlying the findings in their manuscript fully available?

Reviewer #1: Yes

Reviewer #2: Yes

4. Is the manuscript presented in an intelligible fashion and written in standard English?

Reviewer #1: Yes

Reviewer #2: Yes

5. Review Comments to the Author

Reviewer #1: I think that this is a good article. However, I think that introduction should make emphasys on the description of the terms taht the study will evaluate.

Moreover there are different studies related with burnout and empathy in emergency professionals that could be interesting to be referred.

Reviewer #2: Thank you for the opportunity to review this timely and well-written article. More information about the original survey instrument and technique is required. Detailed feedback is as follows:

- Was a sample size calculation performed? If so, please provide.

- Please describe the informed consent process. Where were the participants told the length of time of the survey, which data were stored and where and for how long, who the investigator was, and the purpose of the study?

- If any personal information was collected or stored, describe what mechanisms were used to protect unauthorized access.

- State how the survey was developed, including whether the usability and technical functionality of the electronic questionnaire had been tested before fielding the questionnaire. If it was a previously validated/published survey, then provide the reference.

- Was the survey open or closed? An “open survey” is a survey open for each visitor of a site, while a closed survey is only open to a sample which the investigator knows (password-protected survey).

- How/where was the survey announced or advertised?

- Was the survey web-based or through email? If it is an e-mail survey, were the responses entered manually into a database, or was there an automatic method for capturing responses?

- Were any incentives offered (eg, monetary, prizes, or non-monetary incentives such as an offer to provide the survey results)?

- To prevent biases items can be randomized or alternated. Was this done?

- Did the survey make use of adaptive questioning?

- It is technically possible to do consistency or completeness checks before the questionnaire is submitted. Was this done, and if “yes”, how (usually JAVAScript)? An alternative is to check for completeness after the questionnaire has been submitted (and highlight mandatory items). If this has been done, it should be reported.

- State whether respondents were able to review and change their answers (eg, through a Back button or a Review step which displays a summary of the responses and asks the respondents if they are correct).

- Please define how you determined a unique visitor. There are different techniques available, based on IP addresses or cookies or both.

- What was the view rate? Calculation requires counting unique visitors to the first page of the survey, divided by the number of unique site visitors (not page views!). It is not unusual to have view rates of less than 0.1 % if the survey is voluntary.

- Line 162: Please clarify how response rate was determined. Is this what is typically called the participation rate, or is it the completion rate?

- What was the completion rate? The number of people submitting the last questionnaire page, divided by the number of people who agreed to participate (or submitted the first survey page). This is only relevant if there is a separate “informed consent” page or if the survey goes over several pages. This is a measure for attrition. Note that “completion” can involve leaving questionnaire items blank. This is not a measure for how completely questionnaires were filled in. (If you need a measure for this, use the word “completeness rate”.)

- Registration - In “closed” (non-open) surveys, users need to login first and it is easier to prevent duplicate entries from the same user. Describe how this was done. For example, was the survey never displayed a second time once the user had filled it in, or was the username stored together with the survey results and later eliminated? If the latter, which entries were kept for analysis (eg, the first entry or the most recent)?

- Line 92: please define advanced practice providers as some readers may be unfamiliar with this term.

- Statistical correction - Indicate whether any methods such as weighting of items or propensity scores have been used to adjust for the non-representative sample; if so, please describe the methods.

- How were duplicate surveys identified? Was there an IP check, log file analysis, etc.?

- Line 122: please provide the threshold/definition of patient harm.

- Line 122: please clearly define ‘wrong medication’. For example, did the wrong medication have to actually be administered? What about a wrong order that his caught and changed, for example after inquiry from pharmacy or nursing staff?

- Line 122: please clearly define ‘wrong order’. Did the wrong order have to be executed? What if it was placed and then deleted before execution?

- Line 164: please clarify what made these surveys ‘invalid’.

- Table 1: The data shows a respondent predominance of early-mid career white males working at mod-high volume centers. Does this match regional or national demographics in EM? Please comment in the discussion how this may influence the observed results.

6. PLOS authors have the option to publish the peer review history of their article (what does this mean?). If published, this will include your full peer review and any attached files.

Reviewer #1: No

Reviewer #2: No

---

## [Author Response · Author response to Decision Letter 0]

17 Jul 2020

AUTHORS RESPONSE TO EDITORS AND REVIEWERS

As requested, we have included the original letter and comments with our point by point response in red colored font.

PONE-D-20-11949

Fulfillment, Burnout and Resilience in Emergency Medicine– Correlations and Effects on Patient and Provider Outcomes

PLOS ONE

Dear Dr. Wang,

Thank you for submitting your manuscript to PLOS ONE. After careful consideration, we feel that it has merit but does not fully meet PLOS ONE’s publication criteria as it currently stands. Therefore, we invite you to submit a revised version of the manuscript that addresses the points raised during the review process.

We look forward to receiving your revised manuscript.

Kind regards,

Andrew Carl Miller

Academic Editor

PLOS ONE

Journal Requirements:

Response: Yes, we revised our manuscript with the PLOS ONE style.

Response: Sorry for the confusion. This is a secondary data analysis study based on the data collected via a quality improvement project. This original project was contracted with the WellMD team at Stanford University. Therefore, the study team of this manuscript does not own the copyright of the data, and unable to authorize the upload of all available data to the public due to legal restrictions. The authors of the present study had no special access privileges in accessing the datasets as well. The consent signed by the clinicians taking the survey would preclude us from sharing the raw deidentified data. (See quotes from privacy statement below) However, data might be available upon request to group authorized personnel on a case by case scenario. The contact info is: Dr. Jyothindran, Director of Physician Wellness, Integrative Emergency Services, email: Revathi.jyothindran@bswhealth.org

Quote from the privacy statement:

The database will be stored on a password-protected, encrypted computer system that has limited access and is located in a locked office in a controlled facility. The staff will also delete the Provider Wellness Survey data from the Qualtrics system. Your personally identifiable data associated with the Provider Wellness Survey will be accessible only to the limited staff from the independent survey administrator that IES has appointed for the purposes of collecting and maintaining the data. Survey data will not be shared with your institution or others. Data will be de-identified by the survey administration staff. In addition, results from small work groups may be combined with those from larger groups to protect confidentiality and provide meaningful results. The de-identified data will also be shared with the national Provider Wellness Academic Consortium for purposes of benchmarking and program evaluation. None of your answers will be connected to you.

If you accept this privacy statement, the survey administrator will use your email address to link your responses across multiple years of planned Provider Wellness Survey administrations and to link these survey data to other data, such as participation data collected by IES for its health promotion programs. These linkages will allow IES to examine the relationship between survey variables and program participation, and plan future health promotion programming. Information regarding department/specialty will be used to compare work/life wellness and risks of different groups, and identify which health promotion programming will be most effective for different physician groups. Data collected from the online Provider Wellness Survey will be used for the following additional purposes, as permitted or required by law, including:

1. to provide norms against which sub-groups can be compared

2. to suggest recommendations for future physician health promotion programming and benefits design within IES

3. to identify the work/life wellness and risks of different physician groups and compare these risks and wellness variables with national data

4. to apply for, and/or justify, funding for programs that will help promote wellness among physicians and families participating in those programs

5. to investigate the relationship among variables of physicians' work/life wellness, lifestyle behaviors, knowledge, attitudes, mental and physical well-being, and quality of care metrics

6. potentially , to conduct research approved or deemed exempt by the IRB and approved by IES

When used for these purposes (1 through 6 above), your survey data will be combined with the data of other respondents and potentially, with data collected from other sources. Data will be analyzed and reported in aggregate to groups and individuals within IES, to certain other organizations (e.g., health plans), and potentially, to other audiences for research and publication purposes (with IRB approval or exemption). No personally identifying information will be included in such reports.

By agreeing to this privacy statement, you agree that your data from your partially or fully completed survey may be gathered, stored in the database, and used for the purposes described above.

How You Can Access, Change, or Delete Your Information. If you answer any of the survey questions, your responses will be recorded instantly and stored in Qualtrics. At the end of the survey administration period, your responses will be included in the database even if you have not finished the survey by clicking “submit.” If you would like your data changed or removed from the database, please email the independent survey administrator Dr. Jyothindran, at rjyothindran@ies.healthcare. After a dataset has been de-identified and delivered to IES, the survey administrator will not be able to remove individual respondents’ data from the shared de-identified dataset. This privacy statement will be updated periodically to reflect any material changes to our privacy policy.

3. Please include your tables as part of your main manuscript and remove the individual files. Please note that supplementary tables should be uploaded as separate "supporting information" files.

Response: Yes, we include all takes to the main manuscript and removed the individual files. In addition, we uploaded our supplementary tables separately.

Reviewers' comments:

Reviewer's Responses to Questions

Comments to the Author

1. Is the manuscript technically sound, and do the data support the conclusions?

Reviewer #1: Yes

Reviewer #2: Yes

2. Has the statistical analysis been performed appropriately and rigorously?

Reviewer #1: I Don't Know

Reviewer #2: I Don't Know

3. Have the authors made all data underlying the findings in their manuscript fully available?

Reviewer #1: Yes

Reviewer #2: Yes

4. Is the manuscript presented in an intelligible fashion and written in standard English?

Reviewer #1: Yes

Reviewer #2: Yes

5. Review Comments to the Author

Reviewer #1: I think that this is a good article. However, I think that introduction should make emphasis on the description of the terms that the study will evaluate.

Moreover there are different studies related with burnout and empathy in emergency professionals that could be interesting to be referred.

Response: Yes, we revised our introduction and emphasized on the terms that described in this study (e.g. professional fulfillment, burnout, and personal resilience). Since this study mainly focused on three provider wellness domains (PF, BO, and PR), more emphasis on the relationship among these three domains were addressed. To discuss different studies related with burnout and empathy in emergency professions seem to be deviated from this study focus. In addition, provider wellness can be affected by multiple domains including empathy, which has not been addressed in this study. We think this might better fit for placing to the limitation section. We addressed this in our limitation section as the followings: 

“This study has its limitations. Firstly, provider wellness includes numerous areas and this study only chose three domains for provider wellness measurements, which are limited. More provider wellness domains should be studied in the future.” 

Reviewer #2: Thank you for the opportunity to review this timely and well-written article. More information about the original survey instrument and technique is required. Detailed feedback is as follows:

- Was a sample size calculation performed? If so, please provide.

Response: Integrative Emergency Services (IES) is a physician-owned group mainly providing EM service in the North Texas, USA. We intended to include all providers of our group for this study, therefore, due to epidemiology reason, a sample size calculation was not performed.

- Please describe the informed consent process. Where were the participants told the length of time of the survey, which data were stored and where and for how long, who the investigator was, and the purpose of the study?

Yes. The first page of the survey is the introduction to the survey, including the approximately length of time the survey can be completed. Informed consent was also provided on the second page of the survey, survey was not proceeded if providers declined to participate. The second page details the privacy statement and requires the user to provide consent for 1. Saving email address for future longitudinal projects. 2. Proceeding with the survey. The survey is collected on Qualtrics, but then stored in a database that is password protected and only accessible to The Risk Authority, the original creators of the survey. The information is then deleted off of the Qualtrics system. We revised and added them to the revised manuscript.

Quote from the privacy statement:

The database will be stored on a password-protected, encrypted computer system that has limited access and is located in a locked office in a controlled facility. The staff will also delete the Provider Wellness Survey data from the Qualtrics system. Your personally identifiable data associated with the Provider Wellness Survey will be accessible only to the limited staff from the independent survey administrator that IES has appointed for the purposes of collecting and maintaining the data. Survey data will not be shared with your institution or others. Data will be de-identified by the survey administration staff. In addition, results from small work groups may be combined with those from larger groups to protect confidentiality and provide meaningful results. The de-identified data will also be shared with the national Provider Wellness Academic Consortium for purposes of benchmarking and program evaluation. None of your answers will be connected to you.

If you accept this privacy statement, the survey administrator will use your email address to link your responses across multiple years of planned Provider Wellness Survey administrations and to link these survey data to other data, such as participation data collected by IES for its health promotion programs. These linkages will allow IES to examine the relationship between survey variables and program participation, and plan future health promotion programming. Information regarding department/specialty will be used to compare work/life wellness and risks of different groups, and identify which health promotion programming will be most effective for different physician groups. Data collected from the online Provider Wellness Survey will be used for the following additional purposes, as permitted or required by law, including:

1. to provide norms against which sub-groups can be compared

2. to suggest recommendations for future physician health promotion programming and benefits design within IES

3. to identify the work/life wellness and risks of different physician groups and compare these risks and wellness variables with national data

4. to apply for, and/or justify, funding for programs that will help promote wellness among physicians and families participating in those programs

5. to investigate the relationship among variables of physicians' work/life wellness, lifestyle behaviors, knowledge, attitudes, mental and physical well-being, and quality of care metrics

6. potentially , to conduct research approved or deemed exempt by the IRB and approved by IES

When used for these purposes (1 through 6 above), your survey data will be combined with the data of other respondents and potentially, with data collected from other sources. Data will be analyzed and reported in aggregate to groups and individuals within IES, to certain other organizations (e.g., health plans), and potentially, to other audiences for research and publication purposes (with IRB approval or exemption). No personally identifying information will be included in such reports.

By agreeing to this privacy statement, you agree that your data from your partially or fully completed survey may be gathered, stored in the database, and used for the purposes described above.

How You Can Access, Change, or Delete Your Information. If you answer any of the survey questions, your responses will be recorded instantly and stored in Qualtrics. At the end of the survey administration period, your responses will be included in the database even if you have not finished the survey by clicking “submit.” If you would like your data changed or removed from the database, please email the independent survey administrator Dr. Jyothindran, at rjyothindran@ies.healthcare. After a dataset has been de-identified and delivered to IES, the survey administrator will not be able to remove individual respondents’ data from the shared de-identified dataset. This privacy statement will be updated periodically to reflect any material changes to our privacy policy.

- If any personal information was collected or stored, describe what mechanisms were used to protect unauthorized access.

Response: We use standard mechanisms to protect unauthorized access as the followings: 1) any personal information linked to the data will be stored initially in the files with password protections, only PI can get access to the data; 2) initial data was recoded to generate a master data file, each provider will be assigned to a unique number with the deletion of all personal information; 3) all data will be destroyed 3 years after the completion of this project. 

- State how the survey was developed, including whether the usability and technical functionality of the electronic questionnaire had been tested before fielding the questionnaire. If it was a previously validated/published survey, then provide the reference.

Response: Yes. This survey was a previously published survey. We revised and added the reference.

- Was the survey open or closed? An “open survey” is a survey open for each visitor of a site, while a closed survey is only open to a sample which the investigator knows (password-protected survey).

Response: This is a closed survey. Survey was sent to each provide via email. We revised our method in the manuscript.

- How/where was the survey announced or advertised?

Response: We sent the survey to all our group ED providers and encourage provider to participate but not mandatory.

- Was the survey web-based or through email? If it is an e-mail survey, were the responses entered manually into a database, or was there an automatic method for capturing responses?

Response: An email was sent to the providers with a link to the survey web. Therefore, data was automatically captured. We revised and addressed it in the method section.

- Were any incentives offered (eg, monetary, prizes, or non-monetary incentives such as an offer to provide the survey results)?

Response: No incentives was offered regardless of the participations. 

- To prevent biases items can be randomized or alternated. Was this done?

Response: No, we sent the same items to all the participants.

- Did the survey make use of adaptive questioning?

Response: No, the survey did not use adaptive questions.

- It is technically possible to do consistency or completeness checks before the questionnaire is submitted. Was this done, and if “yes”, how (usually JAVAScript)? An alternative is to check for completeness after the questionnaire has been submitted (and highlight mandatory items). If this has been done, it should be reported.

Response: Unfortunately, no mandatory items required to be answered before the completion of this survey, therefore, no consistency or completeness checks done before the questionnaire is submitted.

- State whether respondents were able to review and change their answers (eg, through a Back button or a Review step which displays a summary of the responses and asks the respondents if they are correct).

Response: Yes, the respondents were able to review and change their answers before the final submission. We revised and added to the method of the manuscript.

- Please define how you determined a unique visitor. There are different techniques available, based on IP addresses or cookies or both.

Response: we defined our unique participant based on email and IP addresses. In addition, we also defined each participant with their email and IP address linked to their general information (such as practice location, years of practice, age, gender, etc.). We revised and addressed in our method section. 

- What was the view rate? Calculation requires counting unique visitors to the first page of the survey, divided by the number of unique site visitors (not page views!). It is not unusual to have view rates of less than 0.1 % if the survey is voluntary.

Response: Unfortunately, we are not able to calculate the view rate of this study.

- Line 162: Please clarify how response rate was determined. Is this what is typically called the participation rate, or is it the completion rate?

Response: We determine the response rate as the following:

Response rate = (the number of providers submitted their survey) / (the number of providers received the initial email). In detail: we sent an initial email to 382 providers, and we received 326 survey reports, therefore, an 85% of response rate was found. 

- What was the completion rate? The number of people submitting the last questionnaire page, divided by the number of people who agreed to participate (or submitted the first survey page). This is only relevant if there is a separate “informed consent” page or if the survey goes over several pages. This is a measure for attrition. Note that “completion” can involve leaving questionnaire items blank. This is not a measure for how completely questionnaires were filled in. (If you need a measure for this, use the word “completeness rate”.)

Response: Yes, we had the completion rate of 89% (289/326, we have 289 providers submitted the last questionnaire page with the final submission button and 37 providers declined participating by clicking decline button on the first page). We revised in our result section of the manuscript.

- Registration - In “closed” (non-open) surveys, users need to login first and it is easier to prevent duplicate entries from the same user. Describe how this was done. For example, was the survey never displayed a second time once the user had filled it in, or was the username stored together with the survey results and later eliminated? If the latter, which entries were kept for analysis (eg, the first entry or the most recent)?

Response: We allowed multiple entrances of the survey during the study period. If duplicated surveys were found, we intended to use the later one. However, such issue was not occurred in our study. We revised and addressed in our method section.

- Line 92: please define advanced practice providers as some readers may be unfamiliar with this term.

Response: Yes, we defined advanced practice providers in more detail in the method section of the manuscript.

- Statistical correction - Indicate whether any methods such as weighting of items or propensity scores have been used to adjust for the non-representative sample; if so, please describe the methods.

Response: Unfortunately, we did not use any methods to adjust for the non-representative sample since approximately 10% of sample are considered invalid with over half were empty surveys.

- How were duplicate surveys identified? Was there an IP check, log file analysis, etc.?

Response: Yes, we checked for duplicate surveys by using both IP check with the combination of participants’ general information provided. We revised in our method section.

- Line 122: please provide the threshold/definition of patient harm.

Response: Patient harm defined as any adverse events occurred that directly linked to the medical errors. We revised in our method section.

- Line 122: please clearly define ‘wrong medication’. For example, did the wrong medication have to actually be administered? What about a wrong order that his caught and changed, for example after inquiry from pharmacy or nursing staff?

Response: Yes, we addressed more on the definitions of “wrong medication” and “wrong order” and revised in the method section.

- Line 122: please clearly define ‘wrong order’. Did the wrong order have to be executed? What if it was placed and then deleted before execution?

Response: Yes, we revised and define clearer on “wrong order” in the method section.

- Line 164: please clarify what made these surveys ‘invalid’.

Response: Yes, we revised and clarified the “invalid” survey in the result section.

- Table 1: The data shows a respondent predominance of early-mid career white males working at mod-high volume centers. Does this match regional or national demographics in EM? Please comment in the discussion how this may influence the observed results.

Response: Thanks for reviewer’s valued comment. The average of EM physicians in US is male, 40-50 age, with nearly 10 years of practicing in ED with moderate volume after the residency graduation. However, our providers tend to be younger with less years of practicing in the ED, which has some differences in comparison to national average. Therefore, our finding might lack of generalizability and require external validations. We realized this limitation and addressed more in the discussion. 

6. PLOS authors have the option to publish the peer review history of their article (what does this mean?). If published, this will include your full peer review and any attached files.

Do you want your identity to be public for this peer review? For information about this choice, including consent withdrawal, please see our Privacy Policy.

Reviewer #1: No

Reviewer #2: No

This electronic transmission and any attached files are intended solely for the person or entity to which they are addressed and may contain information that is privileged, confidential or otherwise protected from disclosure under applicable law. Any review, retransmission, dissemination or other use, including taking any action concerning this information by anyone other than the named recipient, is strictly prohibited. If you are not the intended recipient or have received this communication in error, please immediately notify the sender by return email and delete the original message from your system.

---

## [Decision Letter · Decision Letter 1]

6 Oct 2020

Fulfillment, Burnout and Resilience in Emergency Medicine– Correlations and Effects on Patient and Provider Outcomes

PONE-D-20-11949R1

Dear Dr. Wang,

We’re pleased to inform you that your manuscript has been judged scientifically suitable for publication and will be formally accepted for publication once it meets all outstanding technical requirements.

Kind regards,

Sergio A. Useche, Ph.D.

Academic Editor

PLOS ONE

Additional Editor Comments (optional):

Reviewers' comments:

Reviewer's Responses to Questions

**Comments to the Author**

1. If the authors have adequately addressed your comments raised in a previous round of review and you feel that this manuscript is now acceptable for publication, you may indicate that here to bypass the “Comments to the Author” section, enter your conflict of interest statement in the “Confidential to Editor” section, and submit your "Accept" recommendation.

Reviewer #1: All comments have been addressed

2. Is the manuscript technically sound, and do the data support the conclusions?

Reviewer #1: Yes

3. Has the statistical analysis been performed appropriately and rigorously? 

Reviewer #1: I Don't Know

4. Have the authors made all data underlying the findings in their manuscript fully available?

Reviewer #1: Yes

5. Is the manuscript presented in an intelligible fashion and written in standard English?

Reviewer #1: Yes

6. Review Comments to the Author

Reviewer #1: The authors have addressed all the comments i sent in the previous revision. I think that the paper is better right now. Congratulations

7. PLOS authors have the option to publish the peer review history of their article (what does this mean?). If published, this will include your full peer review and any attached files.

Reviewer #1: No

---

## [Editor Report · Acceptance letter]

9 Oct 2020

PONE-D-20-11949R1 

Fulfillment, Burnout and Resilience in Emergency Medicine– Correlations and Effects on Patient and Provider Outcomes 

Dear Dr. Wang:

I'm pleased to inform you that your manuscript has been deemed suitable for publication in PLOS ONE. Congratulations! Your manuscript is now with our production department. 

Kind regards, 

on behalf of

Dr. Sergio A. Useche 

Academic Editor

PLOS ONE